# RETRIEVAL-ENHANCED CONTRASTIVE VISION-TEXT MODELS

**Ahmet Iscen    Mathilde Caron    Alireza Fathi    Cordelia Schmid**
Google Research

## ABSTRACT

Contrastive image-text models such as CLIP form the building blocks of many
state-of-the-art systems. While they excel at recognizing common generic concepts,
they still struggle on fine-grained entities which are rare, or even absent from the
pre-training dataset. Hence, a key ingredient to their success has been the use
of large-scale curated pre-training data aiming at expanding the set of concepts
that they can memorize during the pre-training stage. In this work, we explore an
alternative to encoding fine-grained knowledge directly into the model's parameters:
we instead train the model to retrieve this knowledge from an external memory.
Specifically, we propose to equip existing vision-text models with the ability to
refine their embedding with cross-modal retrieved information from a memory at
inference time, which greatly improves their zero-shot predictions. Remarkably,
we show that this can be done with a light-weight, single-layer, fusion transformer
on top of a frozen CLIP. Our experiments validate that our **r**etrieval-**e**nhanced
**co**ntrastive (RECO) training improves CLIP performance substantially on several
challenging fine-grained tasks: for example +10.9 on Stanford Cars, +10.2 on
CUB-2011 and +7.3 on the recent OVEN benchmark, where we even outperform
the fine-tuned models on unseen classes.

## 1 INTRODUCTION

In the recent years, we have witnessed a surge in the development of vision-language models highly
adaptable to a broad spectrum of downstream tasks (Jia et al., 2021; Radford et al., 2021; Yu et al.,
2022; Chen et al., 2023; Singh et al., 2022). These models work by pre-training two parallel encoders
using contrastive learning (van den Oord et al., 2018) on large-scale, carefully curated, image-text
data (Radford et al., 2021). These two-tower models learn to encode images and texts into an aligned
latent space which enables appealing capabilities such as zero-shot transfer to different downstream
applications, *e.g.* image classification (Radford et al., 2021), image-text retrieval (Plummer et al.,
2015) or open-world recognition (Minderer et al., 2022; Liang et al., 2022). Although these models
have achieved state-of-the-art results across various generic vision-language benchmarks, we observe
that they tend to struggle on tasks requiring a more fine-grained understanding of visual or textual
entities. Our hypothesis is that this disparity stems from the fact that it is hard to align the image
and text modalities. While every image is metaphorically valued at a thousand words, it is often
paired with a short, sometimes noisy, text that neither exclusively nor comprehensively describes it.
For example, current vision-language models are good at associating images of cars with generic
concepts such as "car", "mechanics" or "road trip", because these are common words paired with car
images, but less at finegrained, instance-level, associations such as the specific brand, series or year
of that car. This might therefore produce poor accuracy for zero-shot fine-grained car classification.

The current path taken by the research community has been to ever scale and curate the pre-training
dataset in the hope of covering more and more image-text associations (Radford et al., 2021; Schuh-
mann et al., 2021; Alayrac et al., 2022; Chen et al., 2023). An orthogonal effort has focused instead on
*memory* or *knowledge*-based approaches (Long et al., 2022; Hu et al., 2022; Gui et al., 2021; Izacard
et al., 2022; Guu et al., 2020b; Liu et al., 2023; Shen et al., 2022). These methods, instead of statically
ingesting and memorizing all the world knowledge into model parameters, propose to rely on the
access to an external source of knowledge. For example, K-Lite (Shen et al., 2022) explores how to
improve vision-text models by enhancing the text captions with more comprehensive text definitions

retrieved from an external dictionary, *i.e.* WordNet (Meyer & Gurevych, 2012) or Wiktionary (Miller, 1998). One caveat that we identify in this approach is that initial captions are augmented within their modality only, hence limiting the potential added-value brought by the retrieved items.

To mitigate this issue, we put forth a retrieval-augmented approach that enhances the alignment between image and text representation. A critical observation of ours is that matching representations within the same modality is a significantly simpler task than matching representations across different modalities. To clarify, we observe that the image representation can be effectively utilized to identify images closely resembling the query image, or the text representation can be used to identify texts closely resembling the query text. However, when crossing modalities, these representations are less successful in identifying suitable matches, such as finding the text with the closest representation to a query image representation. We utilize the inherent strength of learned image and text representations within their respective modalities to aid the alignment across modalities. To improve their compatibility, we convert these unimodal representations into a multi-modal format, as conceptually illustrated in Fig. 1. Utilizing a web-scale corpus of image-text pairs for retrieval, we use image representation as a query to identify the top-k most similar images and incorporate the associated text to create a multi-modal representation. In a parallel manner, given a text representation as a query, we find the top-k most similar texts and integrate the associated images to create a multi-modal representation.

Through this process, we successfully transform the image and text representations into multi-modal versions, which significantly simplifies their alignment. Our approach does not presuppose any downstream knowledge and produces a *single* generic model that can be used effectively across different tasks. We show that our method improves over original CLIP (Radford et al., 2021) or LiT (Zhai et al., 2022) models on 11 challenging fine-grained downstream tasks.

## 2 RELATED WORK

**Vision-text pre-training.** While early works have shown the promise of representation learning from image-text paired data (Zhang et al., 2022; Gomez et al., 2017; Joulin et al., 2016; Desai & Johnson, 2021), recent popular papers such as CLIP (Radford et al., 2021) and ALIGN (Jia et al., 2021) have truly unleashed the potential of contrastive image-text pre-training. This paradigm simply works with two parallel uni-modal encoders that learn to distinguish between aligned and non-aligned image-text pairs through a cross-modal contrastive objective (van den Oord et al., 2018; Miech et al., 2020). Appealing properties of these models are simplicity, scalability and great zero-shot performance (Xian et al., 2018). As a result, vision-text contrastive models now form the basic building blocks of more powerful foundational models, such as CoCa (Yu et al., 2022), Flamingo (Alayrac et al., 2022), FLAVA (Singh et al., 2022), and PaLI (Chen et al., 2023) for example. In our work, we enhance the capabilities of the CLIP model (Radford et al., 2021), by adding a light-weight retrieval module. Nevertheless, our method is not specific to CLIP and can be applied to any vision-text model.

**Knowledge-based vision-text models.** Several works have focused on ways of improving upon different aspects of the contrastive vision-text models, such as their training objectives (Gao et al., 2022; Zhai et al., 2022; Dong et al., 2022) or through scaling (Cherti et al., 2022; Pham et al., 2021). Yet, only little exploration has been done on their combination with memory or knowledge-based techniques (Dwibedi et al., 2021; Banani et al., 2023; Liu et al., 2023; Shen et al., 2022; Fan et al., 2023). REACT (Liu et al., 2023) retrieves image-text pairs from an external memory in order to build a training dataset specialized for a specific downstream task. Unlike REACT (Liu et al., 2023), our work does not require any pre-knowledge about the nature of the downstream task, and is hence applicable in a full zero-shot transfer. Another key difference is that our model can leverage items from the memory at inference time, while REACT uses retrieved items to automatically generate a training set to finetune their model. Closer to our work, K-LITE (Shen et al., 2022) learns vision-text models by leveraging external sources of knowledge (*i.e.* WordNet (Meyer & Gurevych, 2012) or Wiktionary (Miller, 1998)) to complete captions with more descriptive content. Unlike our approach, the retrieved knowledge is uni-modal (e.g. they complement text with more text) and the external memory is not used for the image tower. Also using a knowledge-based approach but for image-only representation learning, NNCLR (Dwibedi et al., 2021) finds the visual nearest-neighbor of each training image from a memory for contrastive learning. LGSimCLR (Banani et al., 2023) uses the language guidance to find most similar visual nearest-neighbor. Unlike our work, NNCLR and LGSimCLR only learn visual representations and use retrieval to enhance their supervision during training but not at inference.

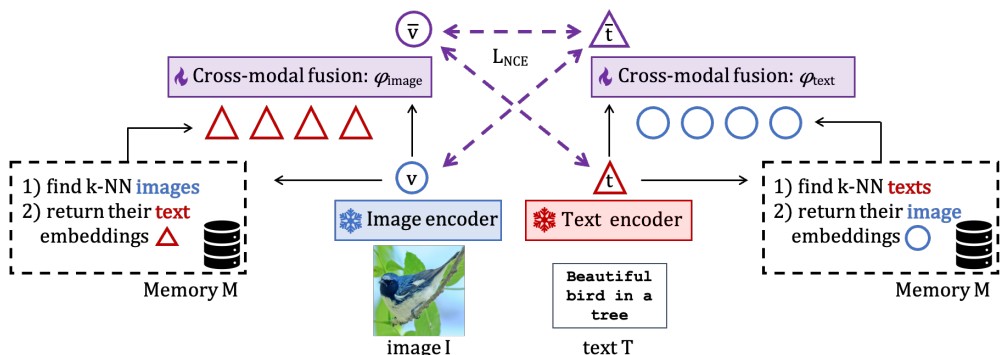

Figure 1: **RECO** works by complementing the frozen representations of pre-trained image-text encoders (such as CLIP) with knowledge retrieved from an external memory. We use an image representation as a query to identify the $k$ most similar images and integrate their associated text embeddings to create a multi-modal representation. Likewise, given a text representation as a query, we find the top-$k$ most similar texts and incorporate their associated images. The fusion of original and retrieved embeddings is done by learning a shallow fusion model to produce improved, multi-modal and knowledge-enhanced versions of the original embeddings. We train for alignment between the refined embeddings, as well as between the refined and original embeddings.

**Retrieval-based methods.** The main argument of the retrieval-based methods is that not all the world knowledge can be compiled into a model's parameters. Thus, the model should also learn to rely on items retrieved from an external memory at inference. Retrieval-based methods have shown their promise in various NLP tasks (Khandelwal et al., 2020; Guu et al., 2020a; Lewis et al., 2020; Wang et al., 2022; Wu et al., 2022; Borgeaud et al., 2022). More recently, there is an increasing interest in the computer vision for retrieval-based methods as well (Blattmann et al., 2022; Fürst et al., 2022; Chen et al., 2022; Long et al., 2022; Hu et al., 2022; Gui et al., 2021; Izacard et al., 2022; Guu et al., 2020b; Liu et al., 2023; Shen et al., 2022; Iscen et al., 2023). Of particular interest, SuS-X (Udandarao et al., 2023) shows that by either retrieving similar samples to the query sample from a large data-bank like LAION can improve zero-shot classification performance of CLIP. Conceptually, SuS-X falls under the *"Cross-modal search and cross-modal fusion"* variant explored in this paper (see second scenario in Fig. 2). RA-CLIP (Xie et al., 2023) enriches the CLIP visual representation by retrieved image and text. However, their attempt to enrich the text representation degrades the performance, whereas we show that the retrieved data can also help produce better text representations.

## 3 METHOD

Our goal is to equip powerful pre-trained vision-language models (such as CLIP) with the ability to complement their representations with cross-modal knowledge retrieved from an external memory. We aim to do this without requiring such models to be retrained from scratch, but by simply learning a light-weight retrieval fusion module on top of them. We emphasize that this work does not propose a new model or loss but rather a new way of adapting pre-trained models to use relevant retrieved knowledge at inference time. An overview of our approach, RECO, is shown in Fig. 1.

**Preliminaries.** We are given a pre-trained frozen dual-encoder vision-text model $f$, where $\mathbf{v} = f_{\text{image}}(I)$ is the embedding of image $I$, and $\mathbf{t} = f_{\text{text}}(T)$ is the embedding of text $T$. We say that these embeddings are *uni-modal* since they are obtained purely from a single modality, either image or text. We assume that image and text embedding spaces are already *aligned*, meaning that they have been trained to produce similar representations for matching image-text pairs and dissimilar representations for non-matching pairs (Radford et al., 2021; Jia et al., 2021; Zhai et al., 2022; van den Oord et al., 2018). This alignment is usually obtained by minimizing the InfoNCE loss (or contrastive loss) (van den Oord et al., 2018) between embeddings of different modalities:

$$\mathcal{L}_{\text{NCE}}(\mathbf{V}, \mathbf{T}) = -\sum_{i=1}^{n} \left[ \log \frac{e^{\mathbf{v}_i^\top \mathbf{t}_i / \tau}}{\sum_j e^{\mathbf{v}_i^\top \mathbf{t}_j / \tau}} + \log \frac{e^{\mathbf{v}_i^\top \mathbf{t}_i / \tau}}{\sum_j e^{\mathbf{v}_j^\top \mathbf{t}_i / \tau}} \right], \tag{1}$$

where $\mathbf{V}$ (resp. $\mathbf{T}$) is the matrix composed of the $n$ visual (resp. text) embeddings in the minibatch and $\tau$ is the temperature parameter. We propose to augment the text and visual embeddings, i.e. $\mathbf{t}$ and $\mathbf{v}$, with external cross-modal knowledge in order to enhance both their expressiveness and their cross-modality alignment. In the following of this section, we first detail how we retrieve relevant

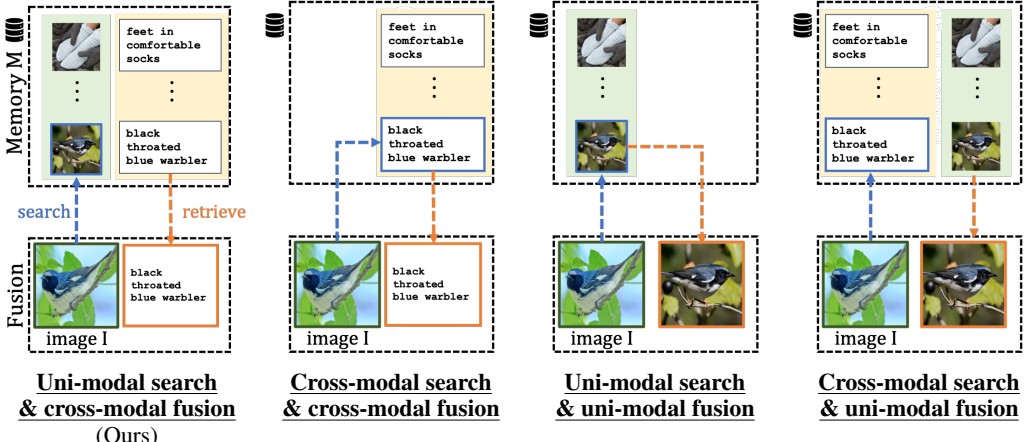

Figure 2: **Conceptual comparison of uni-/cross- modal search and uni-/cross- fusion.** We illustrate the different scenarios for an input image $I$ while the scenarios for text input $T$ are shown in Appendix.

cross-modal knowledge based on within-modality search. Second, we present how we learn to fuse the retrieved information into the original embeddings.

### 3.1 RETRIEVING CROSS-MODAL EXTERNAL KNOWLEDGE

**Memory.** We define the external source of knowledge by a memory $\mathcal{M} = \{(I_i, T_i)\}_{i=1}^{M}$ of $M$ image-text pairs. We assume that $\mathcal{M}$ is very large and covers a broad coverage of concepts. In practice, only a small-subset of $\mathcal{M}$ is relevant for a given input query. Thus, we only consider the $k$ most relevant items from $\mathcal{M}$ for each input obtained by the nearest neighbour search. We denote by $\text{KNN}(\mathbf{v}, \mathcal{M})$ and $\text{KNN}(\mathbf{t}, \mathcal{M})$ the sets formed by the embeddings of the $k$ most relevant items to the queries $\mathbf{v}$ and $\mathbf{t}$ from the memory, where KNN refers to the nearest-neighbour retrieval module.

**Cross-modal fusion.** Our goal is to augment the text and visual original embeddings with cross-modal knowledge, not necessarily learned during the pre-training stage. For example, given the class name *Yellow bellied flycatcher* in a fine-grained bird classification problem such as CUB (Wah et al., 2011), we first look for captions in the memory that are semantically similar to the given class name. We then augment the class name representation with the visual representations of the retrieved similar captions, *i.e.* with what an *Yellow bellied flycatcher* looks like. Likewise, given a visual representation of a bird, we look for similar images in $\mathcal{M}$ and use their corresponding captions in the hope that some of them might contain useful information for our problem such as the species of that bird. Specifically, for a given text or image input, the retrieval module $\text{KNN}(., \mathcal{M})$ returns items with the opposite modality than that of the input. We use the subscripts $v$ or $t$ to specify the modality of the retrieved embeddings. That is, $\text{KNN}_t(\mathbf{v}, \mathcal{M})$ returns text embeddings from an image input and $\text{KNN}_v(\mathbf{t}, \mathcal{M})$ returns image embeddings for text input.

Note that we also evaluate *uni-modal* fusion in our experiments, i.e. complementing visual representations with the retrieved visual knowledge and text representation with the retrieved captions. However, we find in practice that this variant leads to poorer performance than cross-modal fusion, as shown in Tab. 3. Intuitively, we hypothesize that this is because the signal brought by cross-modal fusion is richer due to the complementarity of the different modalities (Iscen et al., 2023).

**Uni-modal search.** We choose to search relevant items in the memory $\mathcal{M}$ based on within-modality similarities, which we refer to as "uni-modal search" as opposed to "cross-modal search". Specifically, we use text-to-text similarity ($t \rightarrow t$) to identify suitable content from a text embedding $\mathbf{t}$ and image-to-image similarity ($v \rightarrow v$) to retrieve relevant matches from a visual embedding $\mathbf{v}$. Formally, let us denote by $\mathbf{V}^{\mathcal{M}}$ and $\mathbf{T}^{\mathcal{M}}$ all the image and text embeddings from $\mathcal{M}$ given by our pretrained vision-text model $f$, *i.e.* we have $\mathbf{V}^{\mathcal{M}} = [f_{\text{image}}(I_1), \ldots, f_{\text{image}}(I_M)]$ and $\mathbf{T}^{\mathcal{M}} = [f_{\text{text}}(T_1), \ldots, f_{\text{text}}(T_M)]$. The retrieval module is hence finally denoted as $\text{KNN}_t^{v \rightarrow v}(\mathbf{v}, \mathcal{M}) = \mathbf{T}^{\mathcal{M}}_{\text{NN}(\mathbf{v}; \mathbf{V}^{\mathcal{M}})}$, *i.e.* for an input image embedding $\mathbf{v}$, the $k$-NN search is done between $\mathbf{v}$ and $\mathbf{V}^{\mathcal{M}}$, but the corresponding $k$-NN indices from the text embeddings $\mathbf{T}^{\mathcal{M}}$ are selected. Similarly, we denote the retrieval process as $\text{KNN}_v^{t \rightarrow t}(\mathbf{t}, \mathcal{M}) = \mathbf{V}^{\mathcal{M}}_{\text{NN}(\mathbf{t}; \mathbf{T}^{\mathcal{M}})}$ for an input text embedding $\mathbf{t}$.

We also evaluate cross-modal search but find that this leads to much poorer performance, especially in fine-grained problems, as shown in Tab. 3. An explanation is that the uni-modal search is an easier task, hence the retrieved elements are more relevant (because more similar) to the input. On the other hand, cross-modal search suffers from the pre-trained CLIP model's lack of fine-grained alignment between the different modalities, resulting in noisier retrieval. Note that another advantage of uni-*versus* cross- modal search is that the latter requires the pre-trained image and text encoders to be already aligned while we can potentially let go of this hypothesis with uni-modal search.

## 3.2 LEARNING HOW TO FUSE THE RETRIEVED KNOWLEDGE

Our goal is to refine the original image and text embeddings $\mathbf{v}$ and $\mathbf{t}$ with the cross-modal knowledge gathered from $\mathcal{M}$. We denote these refined image and text embeddings by $\overline{\mathbf{v}}$ and $\overline{\mathbf{t}}$, defined as $\overline{\mathbf{v}} = \phi_{\text{image}}(\mathbf{v}, \text{KNN}_t^{v \to v}(\mathbf{v}, \mathcal{M}))$ and $\overline{\mathbf{t}} = \phi_{\text{text}}(\mathbf{t}, \text{KNN}_v^{t \to t}(\mathbf{t}, \mathcal{M}))$, where $\phi$ is the *fusion model*.

**Transformer fusion.** We model $\phi_{\text{image}}$ and $\phi_{\text{text}}$ as one-layer multi-head self-attention transformer encoders (Vaswani et al., 2017; Dosovitskiy et al., 2021). Intuitively, this choice allows the original embedding to attend to all the retrieved elements in the fusion process. Note that while the fusion models for text and image encoders have identical architectures, they do not share parameters. In practice, the fusion module has a total of 3.16M parameters, which corresponds to only 2% of the total parameter count when using CLIP-B/32 as the backbone $f$. We have experimented with bigger fusion modules (see Appendix) but find that this light-weight solution works well in practice. We have also tried mean fusion of retrieved and original elements by simply averaging their embeddings but have found in practice that it performs poorly (see Tab. 3). Intuitively, the model needs to learn how to incorporate this new information, by, for example, learning how to omit or enhance some of the retrieved elements.

**Learning.** We train the fusion model $\phi$ on a dataset $\mathcal{D} = \{(I_i, T_i)\}_{i=1}^{N}$ by performing retrieval at training time from the memory $\mathcal{M}$. The pre-trained encoder $f$ is kept frozen. We minimize the alignment loss between the refined embeddings which formally amounts to minimizing the InfoNCE loss of Eq. (1) with the refined embeddings instead of original embeddings, *i.e.* minimizing $\mathcal{L}_{\text{NCE}}(\overline{\mathbf{V}}, \overline{\mathbf{T}})$. We find that it is also sometimes beneficial to perform retrieval for only one of the branches (text or image) at inference time depending on the nature of the downstream task (see Tab. 4). Therefore, we also align the original and refined embeddings by minimizing the following "cross" loss terms: $\mathcal{L}_{\text{NCE}}(\mathbf{V}, \overline{\mathbf{T}})$ and $\mathcal{L}_{\text{NCE}}(\overline{\mathbf{V}}, \mathbf{T})$. This allows to disable one of branches at inference time, since refined and original embeddings are now also aligned. Overall, we minimize:

$$\mathcal{L} = \mathcal{L}_{\text{NCE}}(\overline{\mathbf{V}}, \overline{\mathbf{T}}) + \mathcal{L}_{\text{NCE}}(\overline{\mathbf{V}}, \mathbf{T}) + \mathcal{L}_{\text{NCE}}(\mathbf{V}, \overline{\mathbf{T}}). \tag{2}$$

## 4 EXPERIMENTS

### 4.1 EXPERIMENTAL SETUP

**Training details.** We train the fusion model on top of a frozen CLIP (-B/32 or -L/14 version) model (Radford et al., 2021). We also present a variant of RECO on top of a frozen LiT-L16L (Zhai et al., 2022) model. We train on Conceptual Captions 12M ("CC$_{12M}$") (Changpinyo et al., 2021), an image-text dataset containing about 10M pairs. We use a batch size of 4096, learning rate of $1e^{-3}$ decayed with a cosine schedule and weight decay of $1e^{-5}$. The temperature parameter is learned (Radford et al., 2021). Training is done for 10 epochs, which lasts about 10 hours on a $4x4$ TPUv2 pod. For the memory, we use the subset of WebLI (Chen et al., 2023) containing 1B image-text pairs. We remove the near-duplicates of the test images from the memory. We have also explored using smaller but publicly available memory such as LAION-400M dataset (Schuhmann et al., 2021) and show the results in Appendix.

**Evaluation datasets.** We consider the following six image classification datasets: Stanford Cars ("Cars") (Krause et al., 2013), CUB-200-2011 ("CUB") (Wah et al., 2011), Oxford Flowers ("Flowers") (Nilsback & Zisserman, 2008), ImageNet-1k ("Im1k") (Russakovsky et al., 2015), Places365 ("Pl365") (Zhou et al., 2017) and Stanford Dogs ("Dogs") (Khosla et al., 2011). We also consider the recent Open-domain visual entity recognition (OVEN) benchmark (Hu et al., 2023), containing 729K test images possibly belonging to 6M entity candidates. Finally, we also report performance on text-to-image ("T→I") and image-to-text ("I→T") retrieval on Flickr30k ("Flickr") (Plummer et al., 2015) and MS COCO ("COCO") (Lin et al., 2014) in Appendix. More details about these datasets can be found in Appendix or in their corresponding publication.

Table 1: **Zero-shot transfer to image classification.** We report top-1 accuracy for classification. We show the improvements obtained with RECO on top of CLIP-R50, CLIP-B/32, CLIP-L/14 and LiT-L16L: *absolute* performance gains are between brackets. For reference, we also include the performance of K-Lite (Shen et al., 2022) and RA-CLIP (Xie et al., 2023) (other retrieval-augmented methods) and other image-text models (Align-base (Jia et al., 2021) and PaLI-17B (Chen et al., 2023)). We also report the total parameter count ("# par.") of the different models (in Million).

| Method | # par. | Cars | CUB | Flowers | Im1k | Pl365 | Dogs |
|---|---|---|---|---|---|---|---|
| CLIP-R-50 | 102 | 38.6 | 52.0 | 47.2 | 59.2 | 53.1 | 60.6 |
| + RECO | 114 | 39.8(+1.2) | 62.8(+10.8) | 56.2(+9.0) | 59.4(+0.2) | 54.0(+0.9) | 64.4(+3.8) |
| CLIP-B/32 | 151 | 57.2 | 52.8 | 62.1 | 63.5 | 40.6 | 58.6 |
| + RECO | 154 | 68.1(+10.9) | 63.0(+10.2) | 67.9(+5.8) | 64.6(+1.1) | 42.2(+1.6) | 59.7(+1.1) |
| CLIP-L/14 | 428 | 75.6 | 61.7 | 75.6 | 75.5 | 42.0 | 72.7 |
| + RECO | 435 | 82.8(+7.2) | 73.4(+11.7) | 79.5(+3.9) | 76.1(+0.6) | 43.6(+1.6) | 73.9(+1.2) |
| LiT-L16L | 638 | 90.5 | 54.5 | 77.4 | 80.2 | 45.2 | 75.7 |
| + RECO | 652 | **90.8**(+0.3) | **74.8**(+20.3) | **84.1**(+6.7) | **80.9**(+0.7) | **45.4**(+0.2) | **81.3**(+5.8) |
| *Other approaches* | | | | | | | |
| K-Lite | 151 | 10.0 | – | 78.6 | 52.3 | – | – |
| RA-CLIP | 151 | – | – | – | 53.5 | – | 26.1 |
| Align | 247 | 78.7 | 38.2 | 64.9 | 67.6 | 44.0 | 56.3 |
| PaLI | 17,000 | – | – | – | 72.1 | – | – |

Table 2: **Zero-shot performance on OVEN.** We report top-1 accuracy on seen and unseen categories and their harmonic mean. We also indicate the total number of parameters of each model ("# params").

| Method | # params (M) | Seen | Unseen | Harmonic mean |
|---|---|---|---|---|
| *Zero-shot* | | | | |
| PaLI-17B (Chen et al., 2023) | 17,000 | 4.4 | 1.2 | 1.9 |
| CLIP-L/14 (Radford et al., 2021) | 428 | 5.6 | 4.9 | 5.3 |
| CLIP-L/14 (Radford et al., 2021) + RECO (Ours) | 435 | **11.5** (+5.9) | **13.3** (+8.4) | **12.3** (+7.0) |
| *Fine-tuning on the OVEN Seen categories* | | | | |
| CLIP-L/14 Fusion (Hu et al., 2023) | 880 | 33.6 | 4.8 | 8.4 |
| PaLI-3B (Chen et al., 2023) | 3,000 | 19.1 | 6.0 | 9.3 |
| CLIP-L/14 CLIP2CLIP (Hu et al., 2023) | 860 | 12.6 | 10.5 | 11.5 |
| PaLI-17B (Chen et al., 2023) | 17,000 | 28.3 | 11.2 | 16.1 |

**Evaluation protocol.** We evaluate in the zero-shot setting for all the considered benchmarks, meaning that no adaptation is done to the downstream task. As common in the literature (Radford et al., 2021; Jia et al., 2021; Singh et al., 2022; Zhai et al., 2022), we add prompts to the text of the downstream tasks, following (Zhai et al., 2022). All evaluation protocols are in Appendix.

## 4.2 ZERO-SHOT TRANSFER

**Image classification.** In Tab. 1, we observe that RECO boosts the zero-shot performance of CLIP and LiT on zero-shot image classification with large improvements especially on the fine-grained datasets. For example, we improve the original CLIP-B/32 accuracy by +10.9 on Cars, +10.2 on CUB and +5.8 on Flowers. The performance is also improved on less fine-grained benchmarks such as ImageNet or Places, though by more moderate margins (i.e. respectively +1.1 and +1.6). Secondly, we see in Tab. 1 that the performance gains are consistent across all vision-text backbones (CLIP-R-50, CLIP-B/32, CLIP-L/14, and LiT-L16L). Note that LiT-L16L is pre-trained on Webli, which is our memory bank, and we still observe the benefits of RECO. For reference, we also report in Tab. 1 the numbers from other popular vision-text approaches (Jia et al., 2021; Chen et al., 2023). Overall, the experiment in Tab. 1 confirms our initial motivation that retrieval from an external memory improves zero-shot recognition tasks, especially in fine-grained settings.

**Open-domain visual entity recognition (OVEN).** In Tab. 2, we show the zero-shot performance of RECO on the OVEN benchmark. We see that our method improves greatly over CLIP-L/14 on this challenging task, with an impressive relative improvement of +132%. Note that we do not train or fine-tune our model on the OVEN training set. Remarkably, we observe in Tab. 2 that RECO also

Table 3: **Uni-modal search for cross-modal fusion.** We report top-1 accuracy for zero-shot image classification. We evaluate the impact of uni-modal versus cross-modal search and uni-modal versus cross-modal fusion. These different mechanisms are conceptually illustrated in Fig. 2. We report *absolute* improvement between brackets and the average *relative* improvement over not using retrieval (i.e. CLIP performance) in the last row ("Avg. rel. $\Delta$").

| Search | Fusion | Cars | CUB | Flowers | Im1k | Pl365 | Avg. rel. $\Delta$ |
|--------|--------|------|-----|---------|------|-------|-------------------|
| – | – | 57.2 | 52.8 | 62.1 | 63.5 | 40.6 | – |
| $\phi$ = *Transformer fusion* | | | | | | | |
| 1 Uni-modal | Cross-modal | **68.1** (+10.9) | **63.0** (+10.2) | **67.9** (+5.8) | **64.6** (+1.1) | **42.5** (+1.9) | **+ 9.0 %** |
| 2 Cross-modal | Cross-modal | 56.6 (-0.6) | 53.8 (+1.0) | 64.3 (+2.2) | 64.3 (+0.8) | 42.4 (+1.8) | + 1.7 % |
| 3 Uni-modal | Uni-modal | 57.3 (+0.1) | 51.2 (-1.6) | 62.2 (+0.1) | 62.1 (-1.4) | 41.7 (+1.1) | − 0.4 % |
| 4 Cross-modal | Uni-modal | 54.0 (-3.2) | 50.7 (-2.1) | 61.4 (-0.7) | 62.3 (-1.2) | 41.2 (+0.6) | − 1.9 % |
| $\phi$ = *Mean fusion* | | | | | | | |
| 5 Uni-modal | Cross-modal | 46.9 (-10.3) | 44.9 (-7.9) | 50.5 (-11.6) | 40.1 (-23.4) | 23.7 (-16.9) | − 21.7 % |
| 6 Cross-modal | Cross-modal | 43.7 (-13.5) | 45.3 (-7.5) | 58.7 (-3.4) | 55.2 (-8.3) | 32.7 (-7.9) | − 11.0 % |
| 7 Uni-modal | Uni-modal | 44.0 (-13.2) | 47.2 (-5.6) | 61.3 (-0.8) | 55.1 (-8.4) | 36.2 (-4.4) | − 9.8 % |
| 8 Cross-modal | Uni-modal | 33.4 (-23.8) | 30.2 (-22.6) | 38.9 (-23.2) | 40.0 (-23.5) | 24.7 (-15.9) | − 33.0 % |

significantly outperforms much bigger models which are directly *fine-tuned for this task*, for example CLIP2CLIP (Hu et al., 2023) or PaLI-3B (Chen et al., 2023) while using respectively 2 × and 7 × less parameters. It even comes close to the performance of PaLI-17B while being 39 × smaller and not using any fine-tuning.

### 4.3 DESIGN CHOICE ANALYSES

In this section, we validate several components of our model, namely the uni-modal search and cross-modal fusion, training of the fusion module and the number of retrieved elements from the memory. We also propose some qualitative examples to help understanding why RECO improves over CLIP performance. We use ViT-CLIP-B/32 throughout this section.

**Uni-modal search and cross-modal fusion.** In Tab. 3, we evaluate different alternatives for our method, namely (i) performing cross-modal search in the memory instead of uni-modal search and (ii) fusing uni-modal items (i.e. combining text with text and image with image) instead of cross-modal fusion. These different scenarios (uni- *versus* cross- modal search and fusion) are detailed in Section 3.1 and conceptually illustrated in Fig. 2. Firstly, we observe in Tab. 3 that uni-modal search (row 1) leads to a better performance compared to cross-modal search (row 2), with +9.0 *versus* +1.7 average relative improvement over CLIP. We remark that the gap is especially important for fine-grained datasets such as Cars, CUB and Flowers. This agrees with our hypothesis that cross-modal search suffers from the pre-trained CLIP model's lack of fine-grained alignment between different modalities. By contrast, using the inherent strength of image and text representations within their respective modalities allows to retrieve relevant matches, as qualitatively observed in Fig. 4. Secondly, we observe in Tab. 3 that uni-modal fusion (rows 3 and 4) works substantially worse than cross-modal fusion (rows 1 and 2). Indeed, we see that augmenting text embeddings with other text embeddings and image embeddings with other image embeddings does not bring any significant improvement over the baseline, and even tends to hurt the performance. Intuitively, a possible explanation is that cross-modal fusion allows us to inject complementary signal into the original embeddings (Iscen et al., 2023). By contrast, uni-modal provides signal that is already similar to the input, hence not as much additional information. Finally, we see in Tab. 3 that all the variants (rows 5, 6, 7 and 8) fail when simply averaging retrieved and original embeddings instead of learning the fusion with a transformer. This highlights the importance of *learning* to incorporate the retrieved items to the original embeddings before deploying the model at inference.

**Image and text retrieval fusion modules.** In Tab. 4, we compare models trained to fuse only text original embeddings (row 1), only image original embeddings (row 2) or both (row 3). We observe that while models trained to fuse only image or text perform reasonably well on some benchmarks, they typically lag behind on other benchmarks. For example, the model trained for only image fusion (row 2) is strong on zero-shot Dogs benchmark but behind on CUB and COCO. Secondly, as shown in Tab. 4, unlike the vision-only or text-only variants, our model can be used in different modes at inference time in a flexible manner. Indeed, because we have trained it to align the refined embeddings

Table 4: **Image and text retrieval fusion modules.** We report zero-shot top-1 accuracy for image classification and recall@1 for image retrieval. We compare models trained only for text fusion (row 1), image fusion (row 2) or both (row 3). Our model can be used in different modes at inference: retrieval only for image ($\overline{\mathbf{v}}$), retrieval only for text ($\overline{\mathbf{t}}$) or retrieval for both image and text ($\overline{\mathbf{v}}\&\overline{\mathbf{t}}$).

| | | | CUB | | | | Dogs | | | | COCO T→I | | | |
|---|---|---|---|---|---|---|---|---|---|---|---|---|---|---|
| $\phi_{\text{image}}$ | $\phi_{\text{text}}$ | fusion training loss | $\overline{\mathbf{v}}$ | $\overline{\mathbf{t}}$ | $\overline{\mathbf{v}}\&\overline{\mathbf{t}}$ | Best | $\overline{\mathbf{v}}$ | $\overline{\mathbf{t}}$ | $\overline{\mathbf{v}}\&\overline{\mathbf{t}}$ | Best | $\overline{\mathbf{v}}$ | $\overline{\mathbf{t}}$ | $\overline{\mathbf{v}}\&\overline{\mathbf{t}}$ | Best |
| 1 | ✓ | $\mathcal{L}_{\text{NCE}}(\mathbf{V}, \overline{\mathbf{T}})$ | ✗ | 59.3 | ✗ | **59.3** | ✗ | 59.6 | ✗ | **59.6** | ✗ | 33.3 | ✗ | **33.3** |
| 2 | ✓ | | $\mathcal{L}_{\text{NCE}}(\overline{\mathbf{V}}, \mathbf{T})$ | 59.7 | ✗ | ✗ | **59.7** | 59.2 | ✗ | ✗ | **59.2** | 31.3 | ✗ | ✗ | **31.3** |
| 3 | ✓ | ✓ | $\mathcal{L}_{\text{NCE}}(\overline{\mathbf{V}}, \overline{\mathbf{T}})+\mathcal{L}_{\text{NCE}}(\mathbf{V}, \overline{\mathbf{T}})+\mathcal{L}_{\text{NCE}}(\overline{\mathbf{V}}, \mathbf{T})$ | 60.0 | 58.4 | 63.0 | **63.0** | 59.7 | 59.7 | 59.4 | **59.7** | 31.9 | 33.6 | 31.7 | **33.6** |

| Method | Train data | CUB |
|---|---|---|
| CLIP-B/32 | – | 52.8 |
| + CLIP-style | $CC_{12M}$ | 44.8 |
| + CLIP-style | $CC_{12M} + \mathcal{R}_{\text{Webli}}$ | 46.8 |
| + RECO | $CC_{12M} + \mathcal{R}_{\text{Webli}}$ | **63.0** |

Figure 3: **(left) Disantangling the effect of additional training and RECO. (middle) Effect of updating the memory after training. (right) Effect of the number $k$ of retrieved elements.** We report zero-shot top-1 accuracy on CUB. The CLIP baseline is shown with symbol ★.

with the original ones (see the cross terms in the loss 2), we can choose to disable the retrieval for one of the branches at inference time, depending on the task. Therefore, we compare in Tab. 4 different options at inference time: using retrieval only for the image input, only for the text input or for both of them, denoted respectively by $\overline{\mathbf{v}}$, $\overline{\mathbf{t}}$ and $\overline{\mathbf{v}}\&\overline{\mathbf{t}}$. We observe in Tab. 4 that depending on the nature of the task, one of these options might be preferable over the others. For example, for image classification we see that augmenting the image embeddings with retrieved text has more positive impact than augmenting the text embeddings, though the best of performance is obtained with both. On the other hand, text and image retrievals seem to benefit more from augmenting the text rather than the image side. This intuitively can be explained by the fact that text descriptions in retrieval benchmarks are typically highly specific compared to the class names in image classification and so augmenting with visual examples of what they refer to greatly helps the alignment. We demonstrate qualitative examples of this hypothesis in Appendix. Overall, at inference time, one can choose the best inference mode for a particular downstream task by validation on a held-out set.

**Is the performance boost merely due to additional training?** We replace RECO with an MLP layer of the same capacity initialized from scratch. We train it in a CLIP-style manner on the subset of Webli that we use when training RECO. We denote this subset by $\mathcal{R}_{\text{Webli}}$: it contains the $k = 10$ nearest-neighbors for each CC12M datapoint retrieved from the Webli dataset, and contains 61M examples. We observe in Fig. 3 (left) that training an extra layer on top of CLIP does not bring any gains and even deteriorates its performance. Indeed, CLIP was extensively trained on a large dataset (Radford et al., 2021) and additional training on a relatively small dataset deteriorates the general-purpose property of its representations. Overall, this experiment validates that the performance gains are due to our method and not to training an additional layer on top of CLIP.

**Updating the memory after training.** A clear advantage of the retrieval-based models is that the external memory can be updated with additional, and more contemporary information. We evaluate the effectiveness of RECO when using a larger memory that is not observed during the training. We first create various random subsets of Webli by randomly removing a percentage of data. Then, we train separate RECO models with each Webli subset as its memory. At inference, we evaluate each RECO model either with the subset of memory that it was trained with, or the full Webli memory. Results are shown in Fig. 3 (center). We observe that training and evaluating RECO with only 1% of Webli as the memory does not show improvements compared to the CLIP baseline. However, we observe a significant improvement when evaluating the same model with full Webli memory at inference. This confirms that RECO is capable of utilizing an updated memory without re-training.

**Effect of the number of retrieved elements.** In Fig. 3 (right), we study the effect of the number of retrieved elements in the memory. We evaluate different numbers of $k$-NN during the training and inference time, *i.e.* we train our model with $k$ items from the memory but use $k'$ at inference. We

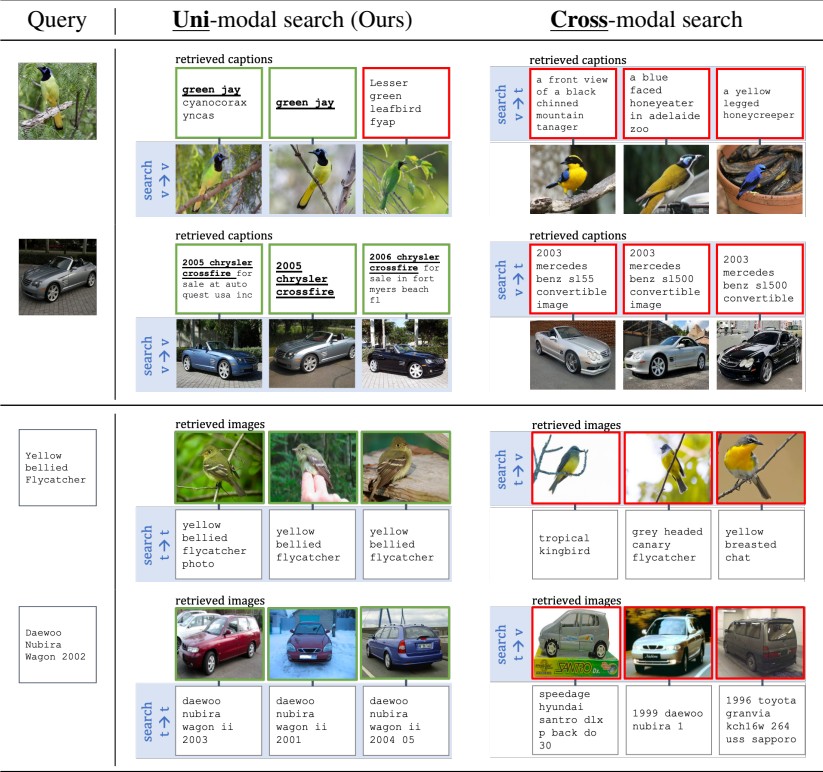

Figure 4: **Qualitative examples on CUB and Cars datasets.** We compare uni- versus cross- modal search for two image queries (top) and two text queries (bottom). Uni-modal search allows to find more suitable matches to the query, which improves the relevancy of the fused elements. We frame in red (resp. green) the unrelevant (resp. relevant) retrieved items to be fused with the query.

see in Fig. 3 (right) that RECO generally obtains a higher performance when $k' > k$ at inference. Interestingly, the performance saturates after $k = 10$. An explanation is that increasing the number of retrieved elements goes with a reduction of the relevancy of the retrieved items.

**Qualitative study.** In Fig. 4, we provide illustrative examples of why RECO can be useful for fine-grained image classification on CUB or Cars datasets. We compare our method with a variant using cross-modal search instead of uni-modal search to illustrate the importance of using the inherent strength of image-only and text-only representations. We observe in Fig. 4 that uni-modal search allows to retrieve better matches for the query. This is because image-to-image or text-to-text search retrieves more similar items to the query than crossing modalities. As a result, retrieved items are more accurate, which leads to a higher accuracy for fine-grained tasks.

**Limitations.** A limitation of this work is that it assumes to have access to a large and rich source of image-text pairs knowledge. While we show in Appendix that public datasets , *e.g.* LAION (Schuhmann et al., 2021), can serve this purpose, the best of performance is obtained with a large private memory. Alternatively, one could use search engine APIs as the memory. Another limitation is that the performance gains of RECO come at the cost of increased inference time. In practice, we use a highly-optimized approximate $k$-NN algorithm (Guo et al., 2020). It takes about 14ms to query Webli (956M examples) with a single 512-d ViT-B/32 CLIP embedding. Using retrieval at inference time incurs an overhead of $25\%$ compared to not using any retrieval at inference time (e.g. baseline CLIP model), but improves the accuracy by up to 10.9.

## 5 CONCLUSION

In this paper, we introduce RECO, a method that enhances the fine-grained recognition capabilities of pre-trained vision-text models. Our approach shows the importance of uni-modal retrieval, yet cross-modal fusion for image and text inputs. We show that RECO consistently improves the performance on 11 zero-shot tasks and that the gains are especially important in challenging fine-grained tasks.

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
