Table 5: **Zero-shot transfer to retrieval.** We report recall@1 for retrieval. We show the improvements obtained with RECO on top of CLIP-B/32 and CLIP-L/14: *absolute* performance gains are between brackets. For reference, we also include the performance of other standard image-text foundation models (Flava (Singh et al., 2022) and Align-base (Jia et al., 2021). We also report the total parameter count ("# par.") of the different models (in Million).

| | | T→I | | I→T | |
|---|---|---|---|---|---|
| Method | # par. | COCO | Flickr | COCO | Flickr |
| CLIP-B/32 | 151 | 30.2 | 61.1 | 51.2 | 80.9 |
| + RECO | 154 | 33.6(+3.4) | 65.7(+4.6) | 52.2(+1.1) | 81.8(+0.9) |
| CLIP-L/14 | 428 | 35.2 | 68.6 | 57.2 | 87.5 |
| + RECO | 435 | 38.7(+3.5) | **72.6**(+4.0) | **58.0**(+0.8) | **88.5**(+1.0) |
| *Other approaches* | | | | | |
| Flava | 172 | 38.4 | 65.2 | 42.7 | 67.7 |
| Align | 247 | **40.2** | **72.6** | 55.1 | 86.7 |

## A   APPENDIX

### A.1   IMAGE/TEXT RETRIEVAL.

Tab. 5 shows that RECO allows to boost the zero-shot performance of CLIP for image and text retrieval even further. We observe that we can improve the recall@1 up to 4.6 points for text-to-image retrieval. We also see smaller, but consistent gains for image-to-text retrieval.

### A.2   USING LAION-400M AS THE MEMORY

In Table 6, we show that our method also works when using a public dataset as the memory bank instead of our private source of knowledge. Indeed, we observe that using LAION-400M (Schuhmann et al., 2021) as the memory bank for RECO gives substantial gains of performance compared to the CLIP baseline across our different zero-shot tasks: for example +6.9 on Cars and +6.1 on CUB. This validates that our method is generic and can work with different choices of external knowledge.

Table 6: **Choice of memory bank.** We report zero-shot top-1 accuracy on different image classification tasks. We evaluate RECO when using two different sources of knowledge: the non publicly available WebLI (Chen et al., 2023) dataset (our default) and the publicly available LAION-400M (Schuhmann et al., 2021) dataset.

| Memory bank | Public | Cars | CUB | Flowers | Im1k | Pl365 |
|---|---|---|---|---|---|---|
| None | – | 57.2 | 52.8 | 62.1 | 63.5 | 40.6 |
| WebLI (Chen et al., 2023) (default) | ✗ | 68.1 (+10.9) | 63.0 (+10.2) | 67.9 (+5.8) | 64.6 (+1.1) | 42.2 (+1.6) |
| LAION-400M (Schuhmann et al., 2021) | ✓ | 64.1 (+6.9) | 58.9 (+6.1) | 63.7 (+1.6) | 63.4 (-0.1) | 42.3 (+1.7) |

### A.3   MORE COMPLEX FUSION MODULE

In Table 7, we experiment with fusion modules of varying sizes. We observe that using a fusion module of one or two layers works comparatively well. However, using larger fusion modules with more layers, *e.g.* four, six or eight, deteriorates the performance. We hypothesize that this is because increasing the capacity of the fusion creates overfitting. Overall, using a single-layer fusion module brings large gains of performance on top of CLIP while being very light-weight to train.

### A.4   END-TO-END FINETUNING VERSUS FROZEN BACKBONE

In Table 8, we evaluate the behavior of RECO when *finetuning* the original encoders at the same time as the fusion module. We observe in Table 8 that the performance is comparable to keeping the encoders frozen as in our default setting. Freezing the encoders has the advantage of requiring less compute resources. In our implementation and using the same hardware, finetuning CLIP-B/32 along

Table 7: **Size of the fusion module.** For each variant, we report the total number of parameters ("# params") in millions and the percentage of the total parameter count which is part of the fusion modules ("% fusion params"). We report zero-shot top-1 accuracy on three image classification tasks.

| # fusion layer | # params (M) | % fusion params | Cars | Flowers | Im1k |
|---|---|---|---|---|---|
| 0 | 151.3 | 0% | 57.2 | 62.1 | 63.5 |
| 1 (default) | 154.4 | 2.0% | **68.1** | 67.9 | **64.6** |
| 2 | 157.6 | 4.0% | 67.8 | **69.1** | 64.3 |
| 4 | 163.9 | 7.7% | 61.9 | 62.8 | 59.8 |
| 6 | 170.2 | 11.1% | 60.6 | 64.5 | 59.6 |
| 8 | 176.5 | 14.3% | 61.1 | 64.4 | 59.6 |

Table 8: **Full finetuning versus frozen encoders.** Both mechanisms produce good performance but freezing the backbone allows for $1.6\times$ faster training. We report zero-shot top-1 accuracy.

| CLIP encoders $f$ | Cars | Flowers | Im1k |
|---|---|---|---|
| Frozen (default) | 68.1 | **67.9** | **64.6** |
| Finetuned | **68.5** | 67.1 | 64.2 |

with the fusion module has a training step $1.6\times$ longer than working with frozen CLIP-B/32 and training only the fusion module.

### A.5 QUALITATIVE EXAMPLES

**Zero-shot retrieval.** In Figure 5, we show some qualitative examples when applying RECO to zero-shot retrieval tasks on COCO dataset. Specifically, we aim to gain an understanding about why the model benefits more from augmenting the text rather than the image side when applied to retrieval downstream tasks (see Table 4 of the main paper). In Figure 5, we look at three different image-text input pairs from the validation set of COCO and display the retrieved captions fused with the query image as well as the retrieved images fused with the query text.

We observe in Figure 5 that the text captions of an image in COCO retrieval task usually focus on one specific aspect of the image (for example the towel in the image of the bathroom or the British flag on the train). We observe that the retrieved images from the input text are likely to also contain this particular aspect of interest and hence match well with the original caption. For example, the retrieved images from the train with a British flag caption (bottom of Figure 5) all contain representations of vehicles with painted British flags, which is more likely to help the alignment with the original input.

On the contrary, the captions retrieved from the original image may focus on another particularity of the image, not mentioned in the original caption. For example, the captions retrieved from the train image contain information about train numbers, train station locations or operators. This information is not useful for this task, because the ground-truth caption focuses on the fact that the train carriage has a British flag on its side. This brings distracting signal instead of helping the alignment. Overall, we think these qualitative examples help us understand why disabling retrieval for the image input and enabling it for the text side results in a better performance in this task.

**Zero-shot image classification.** In Figure 6, we display some qualitative examples of RECO for zero-shot image classification downstream tasks. Specifically, we consider several query images and their class names and show the corresponding elements (captions for query image and images for query class name) retrieved by our model. We observe in Figure 6 that in majority of cases, given a visual input, searching for similar images and retrieving their corresponding captions effectively returns descriptions containing useful information for fine-grained classification problem. For example, the captions retrieved from the cat image at the top of Figure 6 contain the breed of that cat (siamese). Likewise, given the class name "siamese cat", RECO look for similar captions, for example "picture of a siamese cat", and returns their corresponding images. These all contain visual examples of what a siamese cat looks like. Figure 6 show several successful examples of this mechanism and helps giving intuition about why RECO helps for zero-shot image classification.

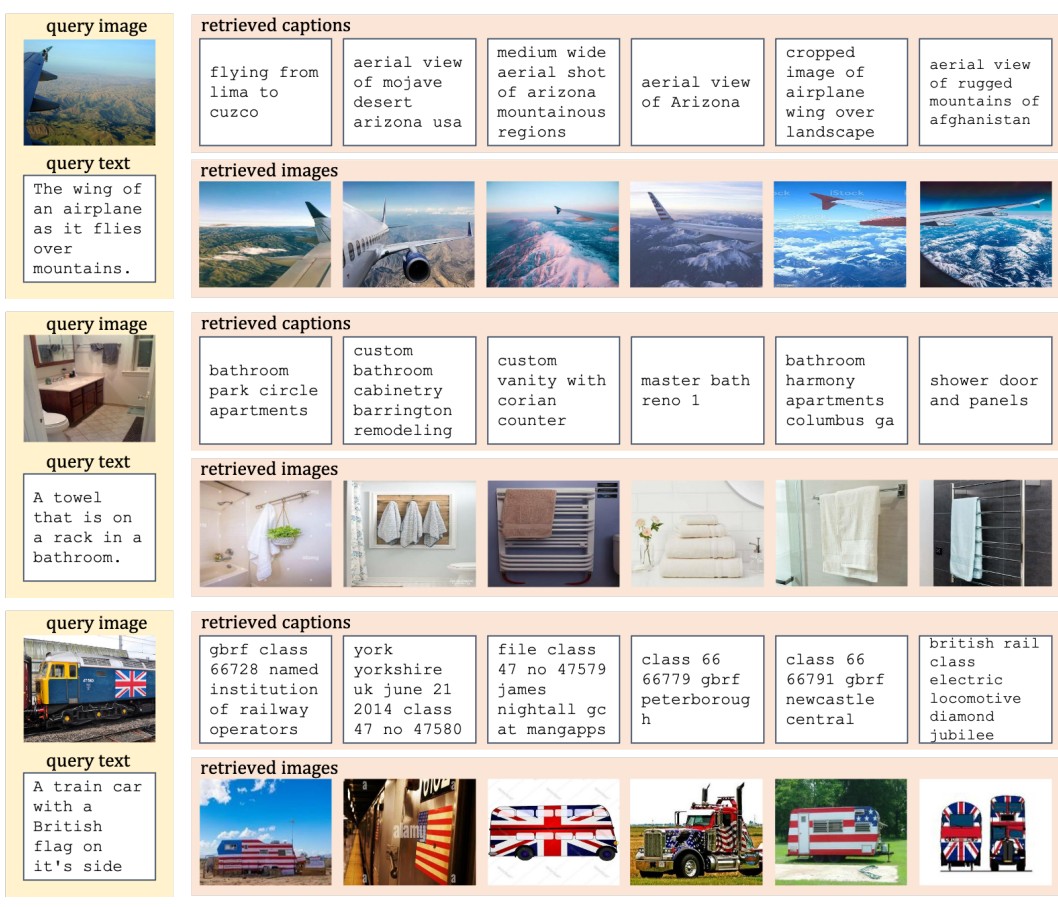

Figure 5: **Qualitative examples of RECO for image and text retrieval.** We display image and text queries on the left panel and retrieved captions and images on the right panel. We observe that retrieved images tend to match better with the input original image than retrieved captions with the input original text. For example, the retrieved captions from the aerial view do not mention a lot "mountains" while this is present in the original text. Instead, they mention many specific locations, for example lima, cuzco, arizona or afghanistan, which are not relevant to the original text description. On the contrary, the retrieved images from the text query are semantically similar to the original image. This qualitatively explains why the best of performance of RECO for zero-shot retrieval is achieved by disabling retrieval on the query image and enabling it on the query text (see Table 4 of the main paper).

Interestingly, we observe some failure cases when retrieving from a query class name which is ambiguous in the sense that it can refer to several things. For example, the retrieved images from the class name "prince of wales feathers" in Flowers dataset returns non useful information such as the emblem of prince of wales or a picture of a feather. This is because "prince of wales feathers" can refer to many things other than a flower species. We observe this behavior for several class names of the Flowers classification benchmark which have a meaning outside of the flower species they refer to; for example "bird of paradise", or "bishop of llandaff" where one of the retrieved image is from the actual person who used to be the Bishop of Llandaff, a community in Wales.

## B EVALUATION DETAILS

### B.1 EVALUATION DATASETS

We report the details of each image classification dataset that we use to evaluate our model. Note that we only use test or validation splits of each of these datasets, training sets are disregarded. Stanford Cars ("Cars") (Krause et al., 2013) contains $8,041$ test images of $196$ fine-grained car classes. Each

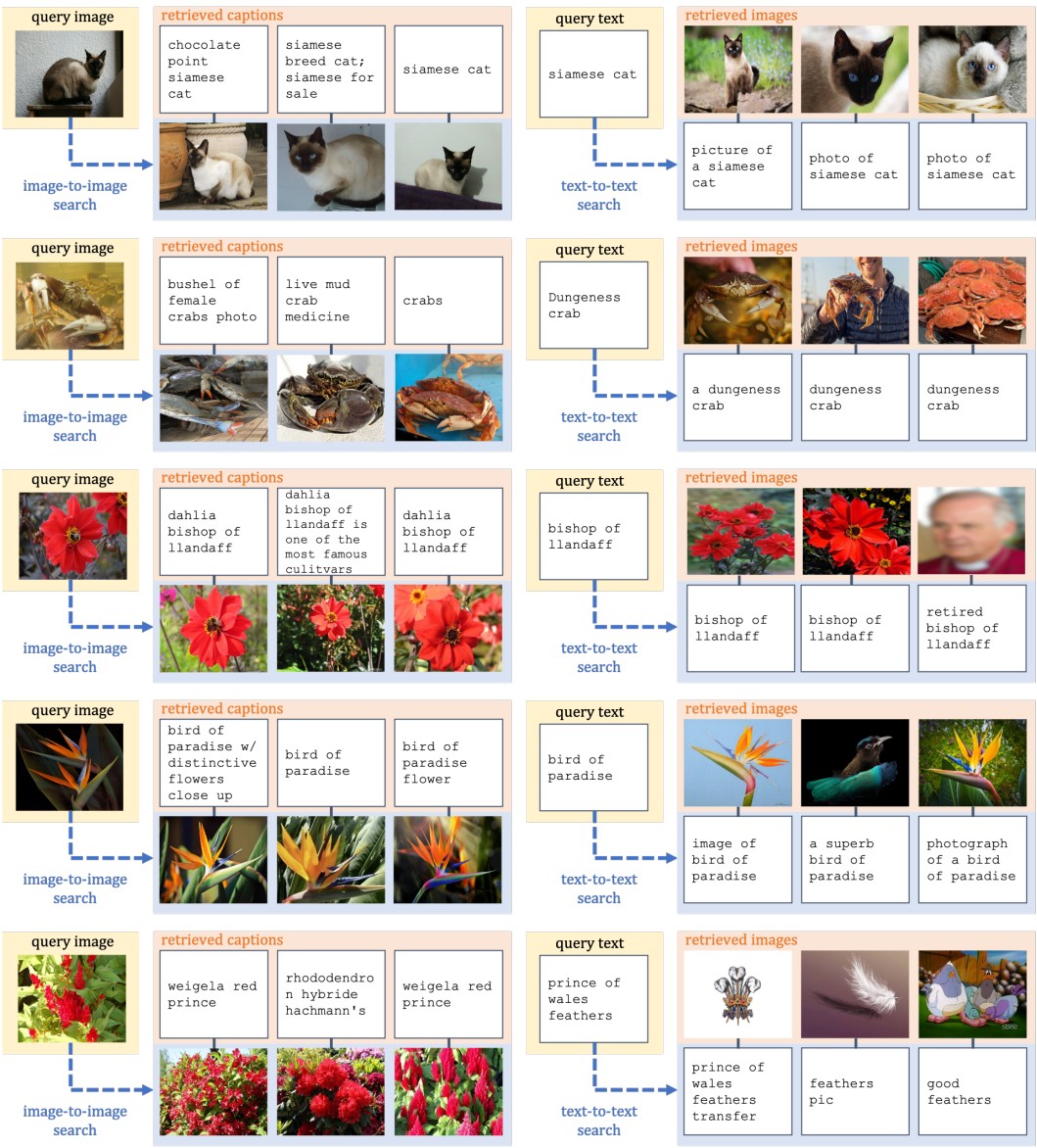

Figure 6: **Qualitative examples of RECO for image classification.** We consider several query images and their class names ("query text") and show the retrieved items to be fused with them. For a given a class name, looking for similar captions and using their corresponding images usually returns relevant visual examples. For example, the images retrieved from the class name "dungeness crab" show examples of what dungeness crabs visually look like. However, when the class name can refer to several things, for example "bird of paradise" which is both a flower and a bird species, then the visual retrieved examples are not always relevant to the finegrained classification problem at hand.

class name consists of make, model and year of a car, *e.g. 2012 Tesla Model S* or *2012 BMW M3 coupe*. CUB-200-2011 ("CUB") (Wah et al., 2011) consists of $5,794$ test images of $200$ bird species. The dataset contains additional annotations, such as part locations, binary attibutes, and bounding boxes, but we do not use any of them. Oxford Flowers ("Flowers") (Nilsback & Zisserman, 2008) has $6,149$ test images of $102$ flower categories commonly found in the United Kingdom. The dataset images contain strong visual variations, such as scale, pose, and light changes. ImageNet-1k ("Im1k") (Russakovsky et al., 2015), or also referred to as ILSVRC 2012, contains $50,000$ validation images from $1,000$ classes. Class names are obtained from the synsets in WordNet (Miller, 1998), and come from a large variety of generic concepts. Places365 ("Pl365") (Zhou et al., 2017) has

Table 9: **Standard deviation of RECO results.** We run five RECO training, each one with a different random seed. We show zero-shot image classification and retrieval results, and their standard deviation across 5 runs.

| Method | Image classification | | | | | | T→I | | I→T | |
|---|---|---|---|---|---|---|---|---|---|---|
| | Cars | CUB | Flowers | Im1k | Pl365 | Dogs | COCO | Flickr | COCO | Flickr |
| CLIP-B/32 | 57.2 | 52.8 | 62.1 | 63.5 | 40.6 | 58.6 | 30.2 | 61.1 | 51.2 | 80.9 |
| RECO | 68.1 ±0.3 | 63.0 ±0.3 | 67.9 ±0.4 | 64.6 ±0.1 | 42.2 ±0.1 | 59.7 ±0.2 | 33.6 ±0.1 | 65.7 ±0.3 | 52.2 ±0.3 | 81.8 ±0.3 |

$36,500$ validation images from 365 generic scene categories. Some examples of class names include *living room, cottage, lecture room, pier etc*. Finally, Stanford Dogs ("Dogs") (Khosla et al., 2011) consists of $8,580$ test images from 120 dog breeds.

We use two datasets for our retrieval experiments. Flickr30k ("Flickr") (Plummer et al., 2015) contains 1000 image-text pairs. Each image contains 5 sentence-level descriptions, or captions. Similarly, MS COCO ("COCO") (Lin et al., 2014) test set, as defined by Karpathy and Li (Karpathy & Fei-Fei, 2015), contains $5,000$ image-text pairs, where each image contains 5 captions. We report the performance for text-to-image ("T→I") and image-to-text ("I→T") retrieval on both datasets.

## B.2 Evaluation protocols

**Zero-shot image classification.** We follow the standard setup (Radford et al., 2021) of embedding each class name with the text encoder. We classify an image to the class which has the highest cosine similarity between the image embedding and the corresping class name embedding. We report top-1 accuracy in all the image classification benchmarks. There is no variance at zero-shot evaluation time since the inference for both text and vision encoders are deterministic.

**Image and text retrieval.** For image-to-text retrieval, given an input image, we rank all the text embeddings according to their similarity to this image embedding. We report the proportion of images that ranks the correct text within the first $R$ positions as the recall@$R$. The process is the symetric for text-to-image retrieval by switching the role of text and image. We report recall@1 in all the retrieval tasks. There is no variance at zero-shot evaluation time since the inference for both text and vision encoders are deterministic.

**Variance and error bars.** We report the performance variance on our small CLIP-B/32 setting to make sure that observed gains are significant. We train RECO with CLIP-B/32 backbone 5 times with different random seeds. We perform the evaluation for each model separately and report the accuracy, averaged over 5 run, with the variance in Table 9. We observe that the standard deviation is small across 5 runs, always below $0.4$ across all benchmarks.

**OVEN benchmark.** OVEN benchmark (Hu et al., 2023) is created by combining 14 existing datasets (ImageNet21k-P (Ridnik et al., 2021; Russakovsky et al., 2015), iNaturalist2017 (Van Horn et al., 2018), Cars196 (Krause et al., 2013), SUN397 (Xiao et al., 2010), Food101 (Bossard et al., 2014), Sports100 (Gerry, 2021), Aircraft (Maji et al., 2013), Oxford Flowers (Nilsback & Zisserman, 2008), Google Landmarks v2 (Weyand et al., 2020), and various VQA (visual question answering) datasets (Goyal et al., 2017; Zhu et al., 2016; Krishna et al., 2017; Marino et al., 2019; Singh et al., 2019; Gerry, 2021)) and grounding their categories to Wikipedia entities. The benchmark consists of two splits. Entity Split measures the image recognition or retrieval capabilities of a model, whereas the Query Split is designed as a VQA task. We focus on the Entity Split in this paper.

The Entity Splits contains training, validation, and test splits. However, since we focus on zero-shot image classification in this paper, we ignore the training and validation splits, and evaluate our model (trained on CC12M as discussed in Sec. 4) directly on the test set. The test set contains $729,259$ examples from $20,549$ entities. Each example belongs to a single entity. Nevertheless, total number of candidate entities during inference is $6,084,494$, *i.e.* there are more than 6M distractor entities at inference.

Note that unlike other image classification datasets, each example is an image-text pair in OVEN. The so-called *intent* text accompanies each image, and clarifies the question at hand, *e.g. what is the model of this vehicle?* Similarly, each entity is also an image-text pair, containing the entity name

and entity image. We simply follow the same protocol as other image classification datasets in this paper, and only consider the example image and entity name.

## C    ILLUSTRATIVE COMPARISON OF UNI-/CROSS- MODAL SEARCH AND UNI-/CROSS- FUSION

We give a conceptual comparison of uni-/cross- modal search and uni-/cross- fusion for an image input $I$ in the paper. We now show in Figure 7 this comparison for a text input $T$.

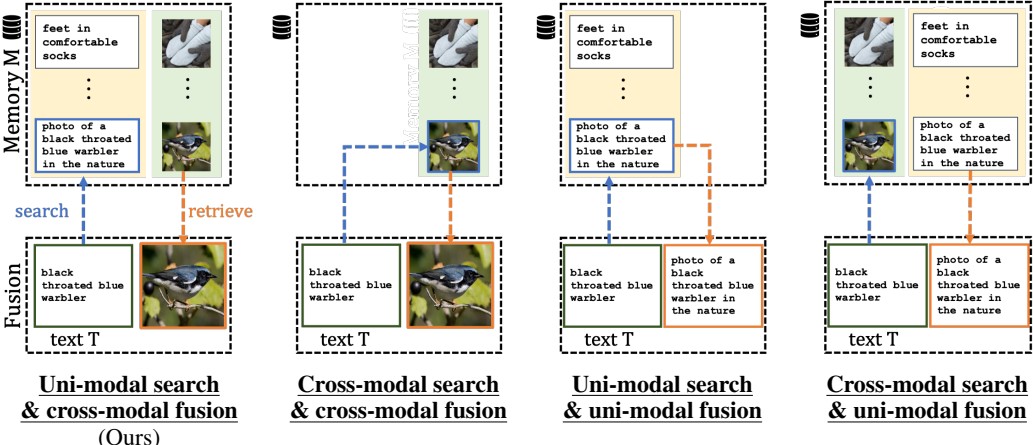

Figure 7: **Conceptual comparison of uni-/cross- modal search and uni-/cross- fusion.** We illustrate the different scenarios for a text input $T$ while the scenarios for image input $I$ are shown in the main paper.

## D    NEAR-DUPLICATE FILTERING

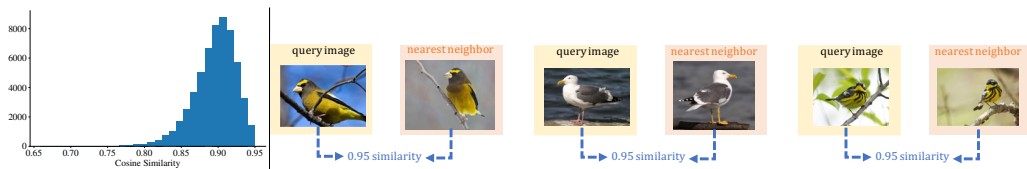

Figure 8: **Left:** Cosine similarity distribution between the test image queries of CUB2011 and their 10 kNN. **Right:** Some of the queries and their nearest neighbors from the memory with 0.95 similarity.

Figure 8 (Left) of the PDF shows the cosine similarity distribution between the queries and memory images on CUB2011 dataset. It is shown that most of the retrieved items have a similarity around 0.9, which indicates that most of the retrieved elements are similar but not identical. Note that there is no cosine similarity above 0.95. This is because we remove any memory image (and their corresponding caption) if they have a similarity of 0.95 or above with any of the test images. Figure 8 (Right) shows some of the kNNs with 0.95 cosine similarity. We see that the retrieved examples are not near-duplicates, but they are conceptually similar as they should be.

## E    BROADER IMPACT

We propose a retrieval-based recognition approach, where we search for similar images and text in a large-scale memory. Data retrieved from such uncurated sources may be biased against certain populations across the world (Prabhu & Birhane, 2020; De Vries et al., 2019). Furthermore, it is important that the privileged user data does not exist in such data collections, in order to avoid using the data without the consent of its owner. We acknowledge these potential misuses, and encourage the community to utilize more fair and responsible data collections.

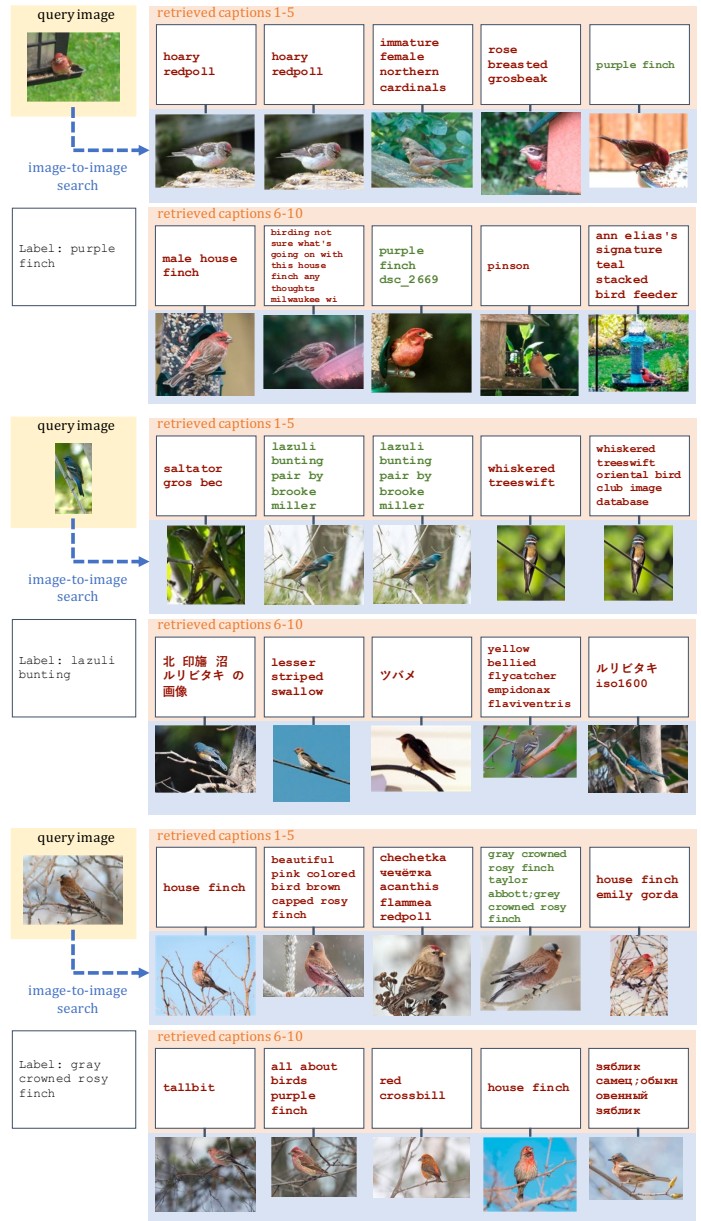

Figure 9: **Noisy examples.** We display image queries and their labels on the left panel and 10 retrieved captions and images on the right panel. Retrieved captions are in green if they contain useful information, red otherwise. RECO learns

## F NOISY EXAMPLES

We show the 10 retrieved captions for some of the test queries on the CUB2011 dataset on Figure 9. We observe that RECO predicts the correct class even if the retrieved captions contain irrelevant examples. This is because the Transformer module learns to aggregate the retrieved captions, implicitly keeping the relevant ones and disregarding the irrelevant ones. Note that Webli contains captions in languages other than English, which are considered noisy for CLIP, because it is trained in an English corpus.