# OpenReview forum: "Retrieval-Enhanced Contrastive Vision-Text Models"
_ICLR.cc/2024/Conference — ICLR 2024 poster_

### Official Review · Reviewer_YY52 · 2023-10-29

**Soundness:** 2 fair
**Presentation:** 2 fair
**Contribution:** 2 fair
**Rating:** 3
**Confidence:** 4

**Summary:**

The paper studies a method to use image-text pretrained model (e.g., CLIP) for fine-grained classification for which CLIP might not have (enough) data. It proposes a method that retrieves relevant data from an external memory, which contains data outside the fine-grained classification dataset. After retrieval, it trains a transformer atop of CLIP to fuse CLIP's features and features of retrieved images. It reports improved fine-grained recognition tasks in experiments.

**Strengths:**

- Retrieval based augmentation is a recent technique for improving performance of downstream tasks using pretrained models.
- Discussion of search methods (e.g., uni-modal search, cross-modal search) is comprehensive.

**Weaknesses:**

There are several concerns related to weaknesses. The paper is hard to follow.


- In Introduction, the paper writes "Our hypothesis is that this disparity stems from the fact that it is hard to align the image and text modalities". It is not clear why it happens w.r.t "it is hard to align the image and text modalities". Can authors clarify?

- Following the above, the abstract mentions that "fine-grained entities which are rare". The first paragraph also uses examples to explain rare concepts. Having concepts rare seems like a different reason from "being hard to align image and text modalities". Which reason is more reasonable? Can authors explain and clarify?

- The sentence is unclear -- "One caveat that we identify in this approach is that initial captions are augmented within their modality only, hence limiting the potential added-value brought by the retrieved items." Can authors clarify?

- The sentence is unclear -- "However, when crossing modalities, these representations are less successful in identifying suitable matches, such as finding the text with the closest representation to a query image representation." Can authors clarify? What message does this sentence deliver?

- The sentence is unclear -- "Through this process, we successfully transform the image and text representations into multi-modal versions, which significantly simplifies their alignment". Can authors clarify?

- The paper studies different search methods as shown in Figure 2. However, it does not discuss computation cost, complexity, etc. It seems that computing features of images and texts can be very computationally expensive. The paper misses important details in methods.

- The paper uses a dataset called WebLI and explains that it is a private dataset. However, how to access the private dataset? Does it mean that authors own the private dataset (indicating a leakage of author identities)? How to fairly compare methods if authors use a private dataset? Authors do not discuss ethical issues w.r.t the private dataset. This is a concern.

- When discussing "Is the performance boost merely due to additional training?", the paper uses ResNet backbone and learns a transformer. Given that transformer might be better than convnets, it is questionable to claim using a transformer is a novel technique. Can authors discuss results if using a transformer backbone in CLIP along with transformer head for fine-grained recognition?

**Questions:**

Questions are in the weaknesses. I encourage the authors to address them in rebuttal.

**Details Of Ethics Concerns:**

The paper uses a private dataset but does not discuss potential ethical issues.

---

> ### Author Response · Authors · 2023-11-22
> **Rebuttal**
>
> **1.  It is not clear why it happens w.r.t "it is hard to align the image and text modalities."**
>
> We explain why we think it is hard to align image and text modalities in the following sentences in the text:
>
> While every image is metaphorically valued at a thousand words, it is often paired with a short, sometimes noisy, text that neither exclusively nor comprehensively describes it. For example, current vision-language models are good at associating images of cars with generic concepts such as “car”, “mechanics” or “road trip”, because these are common words paired with car images, but less at finegrained, instance-level, associations such as the specific brand, series or year of that car...
>
> **2.  Abstract mentions that "fine-grained entities which are rare". Having concepts rare seems like a different reason from "being hard to align image and text modalities". Which reason is more reasonable?**
>
> Both are valid reasons. It is hard to align image and text, and it is especially harder for model to align concepts that are rare.
>
> **3.  "One caveat that we identify in this approach is that initial captions are augmented within their modality only, hence limiting the potential added-value brought by the retrieved items." Can authors clarify?**
>
> Thanks for the comment. We will make sure to make this point clearer. In our paper, we show that it is more beneficial to combine text and image modalities. In K-Lite, the captions are only augmented with additional text and they are not combined with image modality to create cross-modal embeddings.
>
> **4.  The sentence is unclear -- "However, when crossing modalities, these representations are less successful in identifying suitable matches, such as finding the text with the closest representation to a query image representation."**
>
> Thanks for the comment. We will make sure to make this point clearer. Crossing modalities means computing the similarity between images and text, i.e. two different modalities. We show that off-the-shelf CLIP models struggle with this. But it can do a much better job when retrieval is performed within modality (images retrieved given query image, or text retrieved given query text). See cross-modal search in Figure 4 for some examples. In this work, we address this issue by augmenting the input with retrieved images.
>
> **5. The sentence is unclear -- "Through this process, we successfully transform the image and text representations into multi-modal versions, which significantly simplifies their alignment"**
>
> When the query is an image, we augment its embedding with text embeddings corresponding to the similar images. On the other hand, for the text corresponding to the class names, we augment them with the image embeddings corresponding to the similar captions. We perform a similar process when the query is text also. Hence, we end up with multi-modal embeddings. The resulting multi-modal embeddings are easier to align.
>
> **6. The paper does not discuss computation cost, complexity, etc.**
>
> We do discuss the complexity and computation cost in Section 4.3:
>
> Another limitation is that the performance gains of RECO come at the cost of increased inference time. In practice, we use a highly-optimized approximate k-NN algorithm (Guo et al., 2020). It takes about 14ms to query Webli (956M examples) with a single 512-d ViT-B/32 CLIP embedding. Using retrieval at inference time incurs an overhead of 25% compared to not using any retrieval at inference time (e.g. baseline CLIP model), but improves the accuracy by up to 10.9.
>
> **7. The paper uses a dataset called WebLI and explains that it is a private dataset. However, how to access the private dataset? Does it mean that authors own the private dataset (indicating a leakage of author identities)? How to fairly compare methods if authors use a private dataset?**
>
> WebLI is used as the dataset in a series of previous works, including PALI (Chen et al., 2023), PALM-E, CLIPPO etc. Moreover, we show in Table 6 of the Appendix that our method is not specific to WebLI, and we still have large gains when using publicly available LAION dataset.
>
> **8. When discussing "Is the performance boost merely due to additional training?", the paper uses ResNet backbone and learns a transformer. Given that transformer might be better than convnets, it is questionable to claim using a transformer is a novel technique. Can authors discuss results if using a transformer backbone in CLIP along with transformer head for fine-grained recognition?**
>
> The reviewer mentions the section "Is the performance boost merely due to additional training?". We do not use ResNet for results in that section. Instead, we use the ViT-B/32 transformer backbone of CLIP. This is clearly stated in the beginning of Section 4.3: “We use ViT-CLIP-B/32 throughout this section.”
>
> We always make sure that our model uses the same backbone as the baselines we compare against. We will make sure that this becomes more clear in the paper.

---

### Official Review · Reviewer_G4iu · 2023-10-31

**Soundness:** 4 excellent
**Presentation:** 4 excellent
**Contribution:** 3 good
**Rating:** 8
**Confidence:** 3

**Summary:**

This paper proposes RECO, a method to improve the embeddings produced by vision-text encoders like CLIP. For a given image query, the proposed method first finds a set of similar images, with their accompanying texts. Then, the retrieved texts are embedded and fused with the embedding of the query image, to produce a better representation. For text queries, the process is the same, but it first retrieves similar texts and it uses the embeddings of their associated images to improve the embedding of the text query.
The authors evaluate their method on 6 image classification benchmarks, on the OVEN benchmark and on the Text-to-Image and Image-to-Text retrieval tasks from Flickr30k and MS COCO.

**Strengths:**

* The paper presents extensive quantitative results to assess the performance improvements achieved by using RECO on top of different vision-text encoders like CLIP and LiT-L16L, as well as reporting results for different tasks: zero-shot image classification, OVEN task and Text-to-Image and Image-to-Text retrieval. Moreover, the authors report results of strong and adequate baselines. From the results, it is clear that RECO improves CLIP embeddings for all tasks.

* The paper has a strong section on “Design choice analyses”, which does further experiments to show that the specific configuration used by RECO (unimodal search + cross modal fusion) is the best of all the options. Additionally, this section also evaluates the effects of using a different memory bank during inference than the one used during training, the effect of the number of retrieved elements and validates that the improvement does not come from an increased capacity of the model.

**Weaknesses:**

* The main weakness of the paper is that improvement in performance of using RECO changes significantly with the dataset used as the memory bank. The best results are obtained using a non-public dataset (WebLI), for which the authors do not provide any instructions on how to reproduce it.

* Some tables do not report results on the Dogs dataset, it would be better to add them since this dataset is used in the main results Table.

**Questions:**

* Why do the results for Text-to-Image retrieval improve more than Image-to-Text?
* Why is WebLI a better memory bank than LAION? Is the number of images, better alignment between images-text, better captions...?
* Would a model trained with WebLI perform well using LAION during inference?

---

> ### Author Response · Authors · 2023-11-22
> **Rebuttal**
>
> **1.  The best results are obtained using a non-public dataset (WebLI), for which the authors do not provide any instructions on how to reproduce it.**
>
> The details about how the WebLI dataset was collected is described in (Chen et al., 2023) (See Section 3.2, and Appendix B and G of Chen et al., 2023). We further investigate the differences between WebLI and LAION below, in response to other questions from the reviewer.
>
> **2.  Some tables do not report results on the Dogs dataset, it would be better to add them since this dataset is used in the main results Table.**
>
> We add the results for the Dogs dataset from Table 3 below, and will update the paper accordingly:
>
> | Search | Fusion | Accuracy |
> | -------- | -------- | -------- |
> |Uni-modal | Cross-modal | 59.7 |
> |Cross-modal | Cross-modal | 59.2 |
> |Uni-modal | Uni-modal | 59.2 |
> |Cross-modal | Uni-modal | 59.2 |
>
> **3.  Why do the results for Text-to-Image retrieval improve more than Image-to-Text?**
>
> We observe that the baseline CLIP performance is higher for Image-to-Text than Text-to-Image. Therefore, it is easier for our retrieval-augmented approach to provide gains for the text-to-image.
>
> **4.  Why is WebLI a better memory bank than LAION? Is the number of images, better alignment between images-text, better captions...?**
>
> We thank the reviewer for their insightful question. To further examine whether the difference is due to the number of images, or WebLI being higher quality, we randomly subsample 40% of WebLI (to make it the same size as LAION), and re-train our model with this version. We report the results below:
>
> | Memory | Cars | CUB | Flowers | ImageNet1k | Places365 | Dogs |
> | -------- | -------- | -------- | -------- | -------- | -------- | -------- |
> |WebLI - 1B | 68.1 | 63.0 | 67.9 | 64.6 | 42.2 | 59.7 |
> |WebLI - 400M | 66.2 | 61.2 | 65.6 | 64.4 | 42.2 | 59.0 |
> |LAION - 400M | 64.1 | 58.9 | 63.7 | 63.4 | 42.3 | 57.2 |
>
> As shown in the table above, our WebLI-400M results are more comparable with LAION-400M results. The gap in accuracy reduces, but probably WebLI has slightly less noise than LAION.
>
> **5.  Would a model trained with WebLI perform well using LAION during inference?**
>
> We report the results below. We do not observe any meaningful gain in performance compared to a LAION trained model. The trained models learn to fuse the retrieved items in a similar way. Then the quality of the retrieved items at inference play an important factor for the overall performance.
>
> | Memory | Cars | CUB | Flowers | ImageNet1k | Places365 | Dogs |
> | -------- | -------- | -------- | -------- | -------- | -------- | -------- |
> |Webli trained - LAION inference | 64.7 | 59.3 | 63.8 | 63.6 | 42.3 | 57.5 |
> |LAION trained - LAION inference | 64.1 | 58.9 | 63.7 | 63.4 | 42.3 | 57.2 |

---

### Official Review · Reviewer_Smu2 · 2023-11-06

**Soundness:** 3 good
**Presentation:** 2 fair
**Contribution:** 3 good
**Rating:** 6
**Confidence:** 3

**Summary:**

This paper introduces retrieval-enhanced contrastive training (RECO), a method designed to enhance the performance of visual-text models on fine-grained recognition tasks. Specifically, RECO refines the model's embeddings with cross-modal information retrieved from a large external image-text pair dataset. The proposed method outperforms the original CLIP or LiT models in 11 challenging fine-grained tasks.

**Strengths:**

1. The authors have thoroughly investigated various designs for retrieval enhancement, emphasizing the importance of combining uni-modal search and cross-modal fusion.
2. The proposed RECO employs a light-weight, single-layer transformer encoder for fusion, without significantly increasing the number of parameters.
3. They achieve significant improvements on several fine-grained recognition datasets.

**Weaknesses:**

1. This method relies on a large-scale dataset of image-text pairs as external knowledge. However, if the image-text pairs are noisy, the retrieved cross-modal information may be inaccurate, potentially undermining the final performance.
2. The uni-modal search process seems to have a significant overhead (in terms of computation and IO access) during inference, since it has to perform retrieval from a large number of image-text pairs.
3. While this method enhances performance on fine-grained tasks, how does it affect the accuracy of recognizing common generic concepts? Can it be applied to common visual recognition tasks?

**Questions:**

Please see the weakness part.

---

> ### Author Response · Authors · 2023-11-22
> **Rebuttal**
>
> **1. If the image-text pairs are noisy, the retrieved cross-modal information may be inaccurate, potentially undermining the final performance.**
>
> The reviewer is correct in saying that the noisy image-text pairs may undermine the final performance if they are not processed correctly. Our method learns a light-weight transformer, which learns to pool a set of embeddings from the nearest neighbors. The transformer intrinsically learns to filter out irrelevant examples from the kNN list, while keeping the relevant ones. Therefore, we don’t make the assumption that every retrieved item must be relevant.
>
> To study the impact of the noise in our data, and whether the transformer is needed, we design a “Mean fusion” baseline in the paper, which tests whether all the nearest neighbors are already correct, and if we can simply take the average of their embeddings. Table 3 in the paper shows that this is not the case. Mean fusion significantly degrades the performance of the baseline model, which indicates that the nearest neighbors contain more irrelevant examples than the relevant ones.
>
> Finally, we visualize some examples with their retrieved kNNs in the revised PDF (Figure 9 - Appendix). It is shown that RECO predicts the correct class, even if the number of correct images do not outnumber the number of incorrect images in the kNN list.
>
> We thank the reviewer for their suggestion, and will include this discussion in more detail in the future revision.
>
>
> **2. The uni-modal search process seems to have a significant overhead (in terms of computation and IO access) during inference, since it has to perform retrieval from a large number of image-text pairs.**
>
> We use the approximate k-NN algorithm SCANN (Guo et al., 2020)  for retrieval. It takes about 14ms to query Webli (956M examples) with a 512-d embedding to perform retrieval using the SCANN index. Using retrieval at inference time incurs an overhead of 25% compared to not using any retrieval at inference time (note that retrieval runs on CPU and doesn’t use GPU), but improves the accuracy by up to 10.9 points.
>
>
> **3. While this method enhances performance on fine-grained tasks, how does it affect the accuracy of recognizing common generic concepts? Can it be applied to common visual recognition tasks?**
>
> We show the results for ImageNet1k and Places365 as common visual recognition tasks on Table 1. ImageNet1k contains about 60 bird species and about 120 dog breeds, but also “common generic concepts ”, e.g. bookcase, boathouse etc.  Places365 contains generic scene classes, e.g. basement, park, tower etc. In these benchmarks, the performance is still improved, but by more moderate margins (i.e. +1.1 on ImageNet1k and +1.6 on Places365). We hypothesize that this is because the pre-trained CLIP models are already good at recognizing common, generic classes, but don’t perform as well on fine-grained classes (Cars, CUB, Flowers etc).

---

### Author Response · Authors · 2023-11-22
**Rebuttal**

We thank the reviewers for their valuable comments and remarks. The reviewers point out that:
* The paper presents extensive quantitative results, and the authors have thoroughly investigated various designs for retrieval enhancement
* Discussion of search methods and the design choice analyses (e.g., uni-modal search, cross-modal search) is comprehensive.
* From the results, it is clear that RECO improves CLIP embeddings for all tasks.

We address the various concerns and questions by the reviewers in this rebuttal. We address each reviewer’s remarks separately. We also provide a PDF file with more visualizations: Figure 9 (Appendix) of the revised PDF shows the 10 retrieved captions for some of the test queries on the CUB2011 dataset. We observe that RECO predicts the correct class even if the retrieved captions contain irrelevant examples. This is because the Transformer module learns to aggregate the retrieved captions, implicitly keeping the relevant ones and disregarding the irrelevant ones. Note that Webli contains captions in languages other than English, which are considered noisy for CLIP, because it is trained in an English corpus.


If the reviewers have any additional questions, we would be happy to address them.

---

### Meta-Review · Area_Chair_LZYK · 2023-12-05

**Metareview:**

This paper received mixed reviews initially. Two reviewers are positive and one reviewer is negative. The raised issues include unclear technical presentation, unclear motivation explanation, and insufficient experimental validations. During the rebuttal phase, the authors have addressed these issues. The AC monitors the whole process and feels the major issues are solved. The authors shall revise according to these comments in the final version.

**Justification For Why Not Higher Score:**

Retrieval based multi-model contrastive learning is not new.

**Justification For Why Not Lower Score:**

The design is fine and results are promising.

---

### Decision · Program_Chairs · 2024-01-16

Accept (poster)